# Microneedle-Based Delivery: An Overview of Current Applications and Trends

**DOI:** 10.3390/pharmaceutics12060569

**Published:** 2020-06-19

**Authors:** Antonio José Guillot, Ana Sara Cordeiro, Ryan F. Donnelly, M. Carmen Montesinos, Teresa M. Garrigues, Ana Melero

**Affiliations:** 1Department of Pharmacy and Pharmaceutical Technology and Parasitology, Faculty of Pharmacy, University of Valencia, Avda. Vincent Andrés Estellés s/n, 46100 Burjassot, Spain; antonio.guillot@uv.es (A.J.G.); ana.melero@uv.es (A.M.); 2School of Pharmacy, Queen’s University Belfast, Medical Biology Centre, 97 Lisburn Road, Belfast BT9 7BL, UK; a.cordeiro@qub.ac.uk (A.S.C.); r.donnelly@qub.ac.uk (R.F.D.); 3Department of Pharmacology, Faculty of Pharmacy, University of Valencia, Avda. Vincent Andrés Estellés s/n, 46100 Burjassot, Spain; 4Center of Molecular Recognition and Technological Development (IDM), 46100 Burjassot, Spain

**Keywords:** microneedles, microneedle array, transdermal delivery, skin, stratum corneum

## Abstract

Microneedle arrays (MNA) are considered as one of the most promising resources to achieve systemic effects by transdermal delivery of drugs. They are designed as a minimally invasive, painless system which can bypass the *stratum*
*corneum*, overcoming the potential drawbacks of subcutaneous injections and other transdermal delivery systems such as chemical enhancers, nano and microparticles, or physical treatments. As a trendy field in pharmaceutical and biomedical research, its applications are constantly evolving, even though they are based on very well-established techniques. The number of molecules administered by MNA are also increasing, with insulin and vaccines administration being the most investigated. Furthermore, MNA are being used to deliver cells and applied in other organs and tissues like the eyes and buccal mucosae. This review intends to offer a general overview of the current state of MNA research, focusing on the strategies, applications, and types of molecules delivered recently by these systems. In addition, some information about the materials and manufacturing processes is presented and safety data is discussed.

## 1. Introduction

The oral route is the most frequently used for drug administration, due to its simplicity and low cost (specialized staff is not required) [1,2]. In addition, oral solid pharmaceutical forms (tablets, capsules) generally have good physicochemical stability and easy dosing. However, these formulations also face certain obstacles that may impair the bioavailability of the drug such as hepatic first pass effect, complex formation, hydrolysis by gastric pH, gastric and intestinal motility, and drug adsorption and malabsorption due to diseases or surgeries [3,4,5,6]. In addition to these drawbacks, non-adherence to the therapeutic guidelines by the patients is relatively common and makes it harder to achieve drug plasma levels within the therapeutic range [7]. Another factor to consider for oral administration of solid dosage forms is patient acceptance, which can be lower in children and elderly people [8], leading to formulation of oral liquid forms as an alternative. The optimization of these oral pharmaceutical dosage forms has been mostly directed to the prolongation of drug release to reduce the frequency of administration [4]. However, these forms are more complex, from a technological point of view, and any deterioration of them by the patient (i.e., chewing or crushing) or during storage can cause lack of efficacy or even toxicity.

The parenteral administration, considered the natural alternative to the oral route, introduces drugs into the body through a needle, solving some of the aforementioned disadvantages. However, it presents important limitations: Need for aseptic material and techniques, generation of pain, medical complications such as thrombus formation or hypersensitivity reactions, and higher costs due to the administration by trained personnel [9]. The disadvantages and limitations of oral and parental routes triggered the research for other administration sites. In this sense, rectal mucosae, buccal epithelia, or sublingual membrane were explored and demonstrated to be unpredictable, limited, or erratic [10]. Nowadays, a different strategy known as targeting is receiving increasing attention. Targeting enables the drug to directly reach the target organ or tissue, reducing, thus, the systemic exposure and consequently, in many cases, reducing or avoiding toxic side effects [11,12]. In particular, the ophthalmic, respiratory, and cutaneous routes can easily release drugs directly into the corresponding target tissues [13,14].

Administration through the skin [15,16] is another possibility. One of the main assets of this organ is the large surface available, with an area of approximately 20,000 cm^2^ in adults. On the other hand, one of its main functions is to prevent the entry of exogenous substances into the body [17], therefore the skin acts as a barrier for drug absorption. This is mainly achieved by the lipophilicity and the great cohesion between the cells of the most external layer of the skin (*stratum corneum*) [18]. The main process of drug delivery through the skin is passive diffusion, either by intracellular route (the drug is dissolved in the lipidic matrix where the corneocytes are embedded) or by transcellular route (the drug crosses through the cells) [19]. As a passive process, it follows the Fick’s laws, pointing out the importance of the drug concentration in the vehicle and the partition coefficient between vehicle and membrane, among other parameters that condition drug absorption [20]. The diffusion itself depends both on the integrity of the skin and the physicochemical characteristics of the molecules, which must have a low degree of ionization [21], an intermediate distribution coefficient, and a molecular weight of less than 500 Da [22].

Research on cutaneous permeability has shown that, when using a suitable formulation, the skin may also be used for systemic administration of drugs, a process called transdermal drug delivery [23]. This happens because the dermis (one of the deeper layers of skin) is rich in blood microcirculation and when the drug reaches this layer, it is absorbed [24]. The strategies to overcome the difficult permeation through the skin intend to modify the barrier function of the *stratum corneum* or to improve the drug physicochemical properties. They can be classified into chemical or physical methods [25]. The main advantages and disadvantages of these methods are summarized in Table 1.

Microneedle arrays (MNA) are devices that contain microscopic needle-like projections that can perforate the *stratum corneum*, creating ducts that facilitate the flow of large molecules (>500 Da), proteins and nanoparticles through the skin [48]. Different types of MNA have been designed, including solid, coated, hollow, dissolving, and hydrogel-forming microneedles. The microneedle field has been constantly growing since the registration of the first patent (solid and hollow microneedles) by Gastrel and Place in 1971, and shortly after, the second one (coated microneedles) by Paul in 1975. However, we had to wait until the year 2000 for the emergence of dissolving microneedles and the current main applications of MNA, which include gene delivery, vaccination, diagnosis, and cosmetic applications [49]. MNA are a clearly emerging field within the pharmaceutical research, given the rising number of articles being published since the appearance of the “microneedle” concept. In particular, the number of microneedle advertisements was moderate and steady before the year 2000, but afterward, there has been a progressive increase year after year up until today. An autoregressive integrated moving average model (ARIMA) study that extrapolates future trends in the short-medium term based on previous data, identified and predicted a growing trend that is currently being met [50]. The number of patents registered in recent years also shows a continuous growth of MNA as drug delivery systems [51,52,53]. The interest in MNA is related to advantages such as ease of use (can be self-administered and removed) and controlled release of drugs. Moreover, they can be considered a painless transdermal delivery system because the size of microneedles is enough to surpass the first layers of the skin, while avoiding contact with nervous terminations present in the dermis [54].

The aim of this review is to summarize the current knowledge on MNA as transdermal delivery systems, offering an overview regarding different types of MNA, materials and manufacturing methods of these devices, and examples of applications of this technology. MNA application in other routes is also explored. Safety data is also presented, as well as future prospects for this emerging technology.

## 2. Microneedle-Based Transdermal Delivery Approaches

The success of MNA-based drug delivery is determined by critical issues, mainly concerning the design of the MNA (shape, size, geometry, and manufacturing materials and processes) and the type of active substance delivered. The different strategies can be classified as: “*poke and patch*”, “*poke and flow*”, “*coat and poke*”, and “*poke and release*” [55].

### 2.1. Solid Microneedles for “Poke and Patch”

The “*poke and patch”* approach consists in the use of solid MNA to perforate the skin, creating microchannels that reach the deepest layers of the epidermis. This method significantly improves the passive transport of drugs through the skin, since the main barrier to permeability, the *stratum corneum*, is disrupted [56,57]. This approach presents two steps: First, the MNA are used to pierce the epidermis and are subsequently removed; and second, the drug is applied in a conventional dosage form (solution, cream, or patch), which works as an external drug reservoir (Figure 1) [58]. Its simplicity, from the technological point of view, makes it highly attractive, especially for its easy application in a clinical setting. However, this technique is not exempt of controversy and presents several disadvantages. One of the main drawbacks is that the micropores remain open only for a limited time, potentially stopping the delivery of the active substance prematurely. It has been reported that all microneedle treated sites recovered their barrier properties within 2 h [57]. Nevertheless, this period can be extended up to three days under occlusive conditions by using formulations like patches or tapes [59], although the risk of infection increases considerably in these conditions [60]. 

### 2.2. Coated Microneedles for “Coat and Poke”

Another approach with solid MNA is the “*coat and poke”* technique which requires the coating of the solid microneedles’ surface with a drug or vaccine-loaded formulation [61]. This strategy allows drug diffusion from the coating surface to the deeper epidermal layers after MNA insertion (Figure 2) [62]. Certain issues, mainly related to the coating, limit the usefulness of this approach. For instance, the amount of drug which can be encapsulated in the coating layer is relatively low. Besides, the coating’s thickness can decrease the sharpness of the microneedles and influence their ability to perforate the skin [63]. Despite this, coated MNA have shown great efficiency in vaccination, since the antigen dose needed to trigger an immune response is usually in the range of nano or micrograms [64].

### 2.3. Dissolving and Hydrogel-Forming Microneedles for “Poke and Release”.

Dissolving MNA can be made of a range of water-soluble and biodegradable materials in which the drugs can be loaded and released as the MNA dissolves after insertion (Figure 3) [65,66]. The improvement seen in this approach in comparison with the *“poke and patch”* is that dissolving microneedles can maintain controlled drug release over a longer period of time, by controlling the dissolution rate of the formulation used as the MNA matrix. Another advantage is that it reduces the drug administration process to one step, as the MNA are able to pierce the skin and are kept inserted until complete dissolution [67,68]. Besides, dissolving MNA avoids the generation of sharps waste, minimizing the cost related to its management and reducing needle-stick injuries. On the other hand, the drawbacks include a limited drug loading and a potentially lower ability to perforate the *stratum corneum*.

Rapidly separating MNA were designed as a hybrid between coated and dissolving MNA (Figure 4) [69]. The aim is to insert in the skin a drug-loaded water-soluble matrix encapsulating the drug, coupled with a solid MNA composed of an insoluble polymer. This second array helps the insertion of the soft matrix that remains in the skin, while the solid MNA can be easily removed afterwards [70,71]. As an evolution of these MNA, more sophisticated designs have been developed. The insertion of air bubbles in the MNA structure, between the tips and the patch base, enables the easy and rapid separation of the microneedle’s tips from the backing structure after insertion, leaving the tips in the skin and generating non-sharps waste [72].

As an alternative to “poke and patch” approaches, hydrogel-forming MNA or swellable MNA have been developed (Figure 5). The aim of these devices is to imbibe skin interstitial fluid upon insertion to form continuous, unblockable microchannels amongst dermal capillaries. This approach allows the release of less potent drugs contained in an attached patch-type drug reservoir [73,74].

### 2.4. Hollow Microneedles for “Poke and Flow”

The *“poke and flow”* approach was conceived to introduce a drug solution into the skin mimicking hypodermic injections while overcoming their limitations [75,76]. In this approach, the microneedles play a similar role to hypodermic needles, through which drug formulations are administered after skin perforation (Figure 6). Due to their micrometric size, their manufacturing process is difficult and expensive, requiring significant technological resources. By contrast, thanks to the shorter size of these needles, the average patient’s acceptance of this approach is higher than that of traditional injections.

## 3. MNA Fabrication

### 3.1. Materials

MNA are produced using a wide range of materials. All of them must show key properties for the final success of this technology. Any material used for manufacturing MNA should present certain characteristics: Inert nature, absence of immunogenicity, high tensile strength, mechanical strength, low corrosion rate, biocompatibility, and stability. The most common materials for making MNA are metals, silicones, ceramics, glass, sugars, and polymers (Table 2).

### 3.2. Manufacturing Processes

The material or main component of MNA determines the most appropriate microneedle manufacturing technique. In any case, the selected method should be precise, accurate, reproducible, and robust. The most commonly used techniques (listed in Table 2) include laser cutting, laser ablation, electrodeposition, lithography, etching, and micromolding (or solvent casting) [77].

## 4. Applications in Drug and Vaccine Delivery

Parenteral administration is the usual choice for certain drugs such as proteins, antibodies, antigens, and other biotechnological active ingredients [110,111]. These active substances are not adequately absorbed orally, and their molecular weight precludes passive absorption through any alternative route [112]. Additionally, they are particularly sensitive to degradation. Several strategies of encapsulation of these drugs in nano or microparticles have been developed in the last few decades. These molecules are, thus, the most interesting candidates for microneedle-based transdermal delivery. This section summarizes the main types of drugs which have been studied for transdermal administration using MNA and its applications.

### 4.1. Immunization

Transdermal administration is a promising immunization route since the epidermis contains a high number of dendritic and Langerhans cells, which are specialized immune cells. Langerhans cells are able to uptake, process, and present antigens to other immune cells, initiating the immune response [113]. Based on this anatomical fact, different types of MNA, such as dissolving, hollow, and coated, have been used as vaccine delivery systems. A commonly used protein antigen for immunization studies using MNA is ovalbumin (OVA) [114]. Du et al. successfully developed a hollow MNA to administer nanoparticles loaded with OVA and poly(I:C). This research group compared poly (lactic-co-glycolic acid) (PLGA) nanoparticles, liposomes, mesoporous silica nanoparticles, and gelatin nanoparticles, which induced a higher immunoglobulin G2a (IgG2a) antibody response than the OVA/poly(I:C) solution in a murine model. Particularly, PLGA nanoparticles and liposomes, which can control drug delivery, provided a better immunological response [115]. This study shows that hollow MNA are a very interesting tool for nanoparticle-based intradermal vaccination. OVA has also been administered without being encapsulated in nanoparticles. McCrudden et al. designed dissolving MNA made of poly (methyl vinyl ether-co-maleic acid) (PMVEMA) to deliver OVA in a murine model. Results showed that the MNA were able to deliver the active compound, triggering an activation of humoral and cellular responses, which were evidenced by an increase in the production of immunoglobulins and cytokines [116].

Recently, Kim et al. designed MNA containing MERS-CoV-S1 and SARS-CoV-2 vaccines capable of generating potent antigen-specific IgG responses [117]. The trimeric recombinant subunit vaccines against MERS-CoV-S1 and SARS-CoV-2 were produced with and without immune stimulants and were included in dissolving carboxymethyl cellulose (CMC) MNA. Immunogenicity of these vaccine variants delivered either by traditional subcutaneous needle injection or using MNA was tested using an in vivo murine model. MERS-CoV-S1 vaccines delivered by MNA induced stronger humoral responses than traditional needle injections. Besides, the integration of the immune sequences resulted in relatively stronger IgG responses than those without them when vaccines were delivered by subcutaneous injection. MNA delivery of either MERS-CoV-S1 or SARS-CoV-2 induced statistically significant increases in IgG responses compared to pre-immunization groups. In this case, the inclusion of immune stimulants did not cause a significant effect on antibody titer. Furthermore, the immunogenicity of MNA vaccines was maintained after gamma radiation sterilization, ensuring the sterility of the MNA vaccines for clinical applications. Van der Maadden et al. developed hollow microneedles manufactured with hydrofluoric acid fused silica capillaries for the administration of the inactivated poliovirus vaccine (IPV). The intradermal microinjection of 117 ng of viral protein in rats induced an immunological response similar to that elicited by conventional immunization [118]. 

Although most studies with MNA in the vaccination field have focused on viruses, a significant number of studies have also investigated MNA vaccination against bacteria. For example, De Groot et al. immunized mice using hollow MNA loaded with PLGA nanoparticles containing OVA and poly(I:C). The immune response triggered in the murine model, provided protection against a recombinant strain of the intracellular bacterium *Listeria monocytogenes* [119]. Inactivated bacteria as a protective agent have also been formulated in MNA. Rodgers et al. designed a dissolving PMVEMA MNA with heat-inactivated *Pseudomonas aeruginosa*, which were tested using a murine model. Mice were vaccinated using MNA by application of one array to the dorsal surface of each ear and then, they were challenged intranasally with a culture of *Pseudomonas aeruginosa*. Bacterial load in the lungs of mice vaccinated with MNA was significantly lower than those of their unvaccinated counterparts [65]. Similar results were obtained by Liu et al. using specific recombinant protein instead of inactivated bacteria. Staphylococcal enterotoxin B (SEB) protein was formulated in dissolving chondroitin sulfate/trehalose MNA and tested in an in vivo mice model. SEB-specific antibodies were not detected in sera from mice immunized with PBS. In contrast, groups immunized with MNA or intramuscular injections elicited detectable antibody responses after the first immunization. The protective efficacy of MNA was evaluated in the SEB toxin challenge model. Three weeks after the last immunization, the mice were challenged with 1 mg of wild type SEB by intraperitoneal injection and 50 μg LPS after 4 h. All mice vaccinated with a full dose using MNA were completely protected from the toxin challenge, while only 60% of mice from the group treated with the half the dose were protected. In contrast, all mice in the control group died within 24 h after toxin challenge, showing the potential efficacy of the system [120].

DNA plasmid vaccination is used as an effective immunotherapy approach, with the main aim of transfecting the loaded genetic material and vaccination purposes. DNA vaccines are easily produced in large scale and are an interesting option as they show improved stability, safety, and efficacy in comparison with other strategies [121]. Moreover, different studies show that the transdermal delivery of DNA may concede superior protection than conventional intramuscular injections [122,123]. Pamornpathomkul et al. combined a hollow MNA and cationic niosomes loaded with a plasmid DNA encoding OVA (pOVA) to obtain immunization through the skin. The results revealed that this pOVA dosage form successfully induced both humoral and cell-mediated immune responses, including cytokine secretion and IgG, Interleukin-4 (IL-4), and Interferon- γ (IFN-γ) responses compared to a simple administration of naked pOVA. Furthermore, this system also induced a higher level of IgG immune response and cytokine release compared to conventional *stratum corneum* (SC) injections [124]. 

Kines et al. manufactured a human papillomavirus pseudovirus-encapsidated plasmid (HPV16) in coated MNA. Twenty-four hours after the administration of HPV16-Luc pseudo-virions using MNA to mice, samples gave a luminescent signal (continuous for seven days), indicative of an optimal delivery and expression of the luciferase gene. Moreover, serum samples obtained from immunized animals provided evidence that the animals had developed an antibody response, thus proving the efficacy of the designed system [125]. A very similar study was performed by Yan et al., who utilized the plasmid vector pVAX1, which encodes the secreted protein Ag85B of *Mycobacterium tuberculosis*. Specifically, hyaluronic acid (HA) solution (15% w/w) mixed with 8 mg/mL DNA was used to prepare the MNA that were tested in a murine model. The results indicated that this immunization approach significantly enhanced the protective effects generated by a conventional intramuscular administration, measured as IgG1 and IgG2a. Furthermore, a similar result was observed in cellular immune responses by measuring the IFN-γ and Tumor necrosis factor-α (TNF-α) cytokines [126]. 

DNA plasmids are not the only genetic material that can be released by MNA. The delivery of synthetic mRNA is a challenge as new treatments arise based on them for many pathologies, all facing a problem of instability [127]. To overcome this problem, targeted delivery of synthetic mRNA by means of MNA has been explored. Golombek et al. used a porcine ex vivo model to analyze synthetic mRNA-mediated protein expression in the skin after an intradermal delivery of hGLuciferase mRNA by a microneedle injector. Their results evidenced the potential dermal applications of MNA to deliver synthetic mRNAs [128].

### 4.2. Therapy

#### 4.2.1. Therapeutic Proteins

Bovine serum albumin (BSA) is commonly used as a model protein for transdermal drug delivery of therapeutic proteins using microneedles [114]. Cheung et al. used the focused ion beam technique (FIB) to manufacture microneedles and improve the absorption of BSA using the *“poke and patch”* approach. The results of this study revealed that the absorption of BSA through the skin is only possible using a previous perforation of the stratum corneum by means of MNA, in order to create the necessary pathways within the skin to allow the passage of the protein [129]. Different conditions involving a deficit in antibody production are often treated with immunoglobulins, which are administered by injection. Several groups have investigated the possibility of using MNA instead. Mönkäre et al. loaded monoclonal IgG (10% w/w) in dissolving HA MNA. This approach successfully reached the epidermis of ex vivo human skin, and after 10 min, the MNA were almost completely dissolved, depositing the active compound in a depth between 150 and 200 micrometers [130]. In vivo permeation of a high dose of Bevacizumab was studied by Courtenay et al. [131]. The authors demonstrated that the permeation of Bevacizumab with a hydrogel-forming MNA provided an absorption of approximately 25% of the available antibody, while the control solution without microneedles only achieved 0.9%, thus showing a notable enhancement of Bevacizumab permeability by the use of MNA. In addition, in vivo studies confirmed the presence of the drug in plasma for 7 days following a single application, revealing a controlled antibody delivery.

Interferon is a signaling protein used for different therapies, including cancer and infectious diseases. IFNα2b is a subtype of interferon frequently administered subcutaneously or intramuscularly for the treatment of hepatitis B and C infections. Chen et al. showed that the delivery of IFNα2b through the skin was possible using dissolving polyvinylpyrrolidone (PVP) MNA. They observed an in vitro release efficiency of 49% of the dose and a pharmacokinetic profile comparable to the intramuscular injection [132]. Another interesting example is the dissolving CMC MNA developed by Xie et al., which were designed for transdermal delivery of a selective calcitonin gene related peptide (CGRP). The hyperalgesia persisted in the groups treated with blank MNA, while drug loaded MNA were able to induce selective analgesia in different rat pain models, such as spared-nerve injury and diabetic neuropathic pain, without interfering with motor functions and nociception [133].

#### 4.2.2. Insulin

Insulin is probably the most widely investigated hormone to be delivered by an alternative delivery route to the subcutaneous one. Concerning MNA technology, Resnik et al. designed and manufactured hollow silicone MNA to microinject insulin. In vivo results showed a successful infusion of fast-acting insulin determined by monitoring the glucose levels in plasma. Specifically, microneedle-based delivery granted a less significant drop in glucose levels, but a significant increase in serum insulin levels (40–50% of increase), due to a more effective delivery of exogenous insulin [134]. Lee et al. created dissolving MNA composed of gelatin and CMC in a two-step casting and centrifugation process to concentrate the insulin in the needle tip and improve the transdermal delivery efficiency. Ex vivo results showed that 50% of the insulin was released and penetrated the skin after 1 h, with cumulative permeation reaching 80% of the initial dose after 5 h. In vivo results proved a bioavailability of insulin from MNA of 95.6% and 85.7%, respectively [135].

Another example for dissolving MNA in this area is the work performed by Chen et al., who prepared MNA with poly-c-glutamic acid (c-PGA) and polyvinyl alcohol (PVA)/PVP as supporting structures, to be dissolved in 4 min upon skin insertion. The results, obtained in a rat model, indicated that the hypoglycemic effect achieved with insulin-loaded MNA was comparable to the one observed with a conventional subcutaneous insulin injection. The bioavailability of insulin was in the range of 90–97%. Moreover, no significant differences were observed in the plasmatic insulin concentration profiles between serial administrations, demonstrating the stability and accuracy of the MNA [136]. Tong et al. loaded PVP/PVA MNA with glucose and H_2_O_2_ responsive polymeric vesicles containing insulin (Figure 7) [137]. This device is specially designed to deliver insulin under hyperglycemic conditions. Vesicles were made by a self-assembling method and formulated with three polymers: Poly (ethylene glycol), poly (phenylboronic acid), and poly (phenyl boronic acid pinacol ester). Poly (phenylboronic acid) is a glucose-sensitive polymer and poly (phenyl boronic acid pinacol ester) is a H_2_O_2_-sensitive polymer. These polymers are hydrolyzed at hyperglycemic states and in the presence of a H_2_O_2_ stimulus, respectively. Polymeric vesicle contents were coated with glucose oxidase, which is an enzyme that converts glucose into glucuronic acid and then into hydrogen peroxide in the presence of oxygen. Controlled release of the drug from the polymeric vesicles was confirmed by a release nanoparticle study at different glucose and H_2_O_2_ concentrations (200 or 400 mg/dL). The presence of glucose oxidase in the particles provided a faster insulin release since the breakage of the bonds in the poly (phenyl boronic acid pinacol ester) is prompted by the generation of hydrogen peroxide. Then, this group tested the system in a type-2 diabetic animal model. Glucose-sensitive patches were administered to diabetic rats and their glucose blood levels were monitored. MNA loaded with polymeric responsive vesicles were shown as a good alternative to subcutaneous insulin injection, since they achieved an effective and longer-lasting hypoglycemic effect. The system provided a decrease in the baseline glucose level from 500 to 110 mg/dL in 4 h and a gradual recovery to the initial glucose levels during the following 8 h. In contrast, the subcutaneous injection caused a decrease in the baseline glucose levels down to 80 mg/dL in 2 h and a faster recovery to basal levels (in the next 5 h). The decrease in glucose levels (maximum effect) with subcutaneous injections was faster due to the inherent behavior of the patch during release. However, normoglycemic levels (under 200 mg/dL) were maintained for 4 h, in comparison with subcutaneous injection that were maintained for only 2.5 h. 

A similar outcome was achieved using other nanoparticles and patches. Yu et al. designed MNA containing nanovesicles loaded with insulin and glucose oxidase. Glucose responsive nanovesicles were self-assembled from hypoxia-sensitive hyaluronic acid conjugated with 2-nitroimidazole [138]. The local hypoxic microenvironment caused by the enzymatic oxidation of glucose in hydrogen peroxide at the hyperglycemic state is responsible for the dissociation of the vesicles and the patch, allowing the subsequent release of insulin. Similar results were observed using a type-1 diabetic mice model, in which an effective and controlled regulation of blood glucose was achieved. Particularly, a glucose tolerance test was conducted 1 h after administration of the MNA. The control healthy mice exhibited a quick increase in blood glucose level upon an intraperitoneal glucose injection, followed by a gradual decrease to normoglycemia. The diabetic mice treated with MNA which contained insulin and glucose oxidase-loaded particles showed a delayed increase in blood glucose after glucose injection, and then a rapid decline to a normal state within 30 min. However, the glycemia of the mice receiving insulin-loaded MNA did not decline in 120 min, confirming that the MNA which contained insulin and glucose oxidase-loaded particles had significantly faster responsivity towards the glucose challenge.

#### 4.2.3. Vitamins

Many population groups have a considerable risk of suffering vitamin deficiency [139,140,141]. Kim et al. propose a coated MNA loaded with PLGA nanoparticles for supplementation of vitamin D. PLGA nanoparticles containing cholecalciferol were prepared by the emulsion-solvent evaporation method, then coated onto solid metal MNA by dipping them into a 5% (w/v) PVP solution and afterwards into an optimized polymeric formulation containing vitamin D-loaded nanoparticles. These MNA showed a five-fold better delivery performance than a transdermal cream containing chemical permeation enhancers. In particular, the delivery efficiency was 81.08% of the initial dose for the MNA and only 16.28% for the transdermal cream [142].

PLGA nanoparticles loaded with vitamin D were also included in dissolving PVP microneedles by Vora et al. [143]. The release of vitamin D from PLGA nano- and microparticles exhibited a triphasic release profile for up to 5 days. An initial burst release followed by a slower release could be observed. The third release phase was attributed to the broad particle size distribution from nano to micron size of PLGA particles. The ex vivo deposition studies in excised porcine skin showed that the deposition of vitamin D was significantly enhanced by using MNA in comparison with a vitamin D control without microneedles. The drug concentration increased with skin depth, reaching a maximum of 171.7 ± 7.39 μg/cm^2^ at a depth of 0.4 mm. Then, the concentration of vitamin D was progressively halved every 0.2 mm, and at 1.4 mm depth, the drug was no longer detected in significant quantities. In comparison, the control showed almost negligible in-skin drug concentrations. The final cumulative concentration of vitamin D in the skin after MNA insertion was 74.2 ± 9.18% of the initial amount. Similar positive in vitro results were obtained by Hutton et al. in their studies with vitamin K. Using neonatal porcine skin, 35% of the drug content loaded in a dissolving PMVEMA MNA permeated through the skin after 24 h of microneedle insertion [144]. 

In the case of vitamin B_12_, the results with the use of microneedles were especially interesting, since this vitamin has very poor oral absorption. Ramöller et al. developed dissolving PVP MNA with a therapeutically relevant dose of vitamin B_12_. The in vitro delivery of vitamin B_12_ showed that 72.92 ± 5.30% of the total drug load (100 μg) was delivered after 5 h of MNA insertion. The maximum plasma concentration (Cmax) of vitamin B12 after MNA application was 0.37 ± 0.04 μg/mL, while the subcutaneous administration provided a Cmax of 1.30 ± 0.25 μg/mL. No drug was detectable after 24 h in the animals treated with the subcutaneous injection, while the vitamin could be detected in animals treated with MNA after 30 h, proving that MNA provided an extended release [145]. 

#### 4.2.4. Antibiotics

Administration of antibiotics through controlled release drug delivery systems could reduce posology (doses and frequency), contributing to the fight against antibiotic resistance [146]. Compared to parenteral delivery, MNA can represent a big breakthrough in the treatment of infectious diseases. A HA/PVP dissolving MNA containing gentamicin was studied by Gonzalez–Vazquez et al. Results of in vitro permeation of gentamicin showed that 10.25% of gentamicin was released after 3 h, while 75% of the drug was released after 24 h. In vivo studies evidenced that gentamicin plasmatic concentrations after intramuscular injection were higher than those obtained with MNA administration, but the use of MNA allowed more constant plasma concentrations and a slower T_max_. This would mean the possibility to achieve the desired therapeutic effect while spacing the administration frequency [147]. Lee et al. produced bleomycin-coated MNA using a solid MNA and a CMC solution that contained the drug as a coating [148]. The amount of bleomycin delivered by the MNA was measured in an ex vivo essay with porcine skin. Ten minutes after insertion, 74% of the total bleomycin on the microneedles was delivered. When the insertion time increased to 15 min, 82% of the bleomycin was delivered into the skin. After 15 min of insertion, there was no further increase in the amount delivered into the skin, being 82% the maximum amount delivered. Pharmacokinetic profile was obtained using an in vivo rat model comparing the MNA and conventional subcutaneous injections. The bleomycin concentration reached its C_max_ at 50 min and the maximum concentration achieved was 372.18 ng/L. Then, the plasma bleomycin concentration gradually decreased and disappeared within 4 h after administration. The half-life was 36 min for bleomycin from MNA, whereas the half-life by conventional subcutaneous injection was 8 min. 

Mir et al. developed carvacrol (CAR) poly (caprolactone) nanoparticles (NP) as a delivery system and incorporated them in dissolving MNA to achieve a sustained antimicrobial effect at infection sites in wounds [149]. The results showed that the concentration of CAR in dermal layers was considerably higher using MNA with CAR nanoparticles (CAR/NP-MNA) as compared to MNA loaded with free-CAR (CAR-MNA). The dermato-kinetic profile of CAR-MNA depicted that the C_max_ in the epidermis was 73.22 ± 5.48 μg/cm^3^ after 1.42 ± 0.46 h, whereas in the dermis the C_max_ of 116.61 ± 14.61 μg/cm^3^ was achieved in 1.59 ± 0.39 h. In comparison, for CAR/NPs-MNs, the maximum concentration achieved in the epidermis was of 59.15 ± 10.39 μg/cm^3^ at 1.45 ± 0.67 h. In the dermis, the C_max_ of 110.67 ± 19.76 μg/cm^3^ was achieved in 1.77 ± 0.71 h. An ex vivo porcine skin wound infection model demonstrated a significant reduction in the microbial burden in treated groups in comparison to untreated ones. Both MNA showed a reduction in microbial burden in wounds infected by different *S. aureus* strains higher than 90%. In contrast, for the *P. aeruginosa* strains tested, the CAR-MNs showed only 40% of reduction at maximum, while CAR/ NP-MNA showed a minimum of 87% reduction. 

Antimicrobial efficacy of MNA has also been tested using in vitro biofilm wound infection models. Xu et al. developed a dissolvable PVP MNA loaded with chloramphenicol (CHL)-sensitive gelatin nanoparticles. The bacteria chosen to be loaded in the biofilms was *V. vulnificus*, which is usually present in wounds and is able to produce the enzyme gelatinase. In response to the gelatinase produced by the active bacteria, CHL-loaded gelatin nanoparticles are disassembled and CHL is release into the active regions of the biofilm. Toxicity of loaded gelatin nanoparticles were also tested against NIH 3T3 fibroblasts. Comparing to the control group, viable 3T3 cells for the free CHL treated group were only 77.3% of the initial number after 24 h, and 57.2% after 48 h. However, the toxicity of CHL was significantly reduced to half when the drug was encapsulated in nanoparticles, exhibiting a minimal off-target toxicity compared to direct CHL administration. The therapeutic efficacy of the MNA loaded nanoparticles was tested through the viability of bacterial colonies after treating the biofilms with MNA, using equivalent free CHL concentrations as a control. After the incubation of biofilms, viable bacteria were recovered and counted. A significant decrease of 55.6% after 4 h, and 63.2% after 8 h was observed in the colony-forming units per milliliter for the drug-loaded nanoparticles MNA in comparison to the free CHL in solution, showing its effectiveness in treating *V. vulnificus* biofilms in comparison with the drug in free solution [150].

#### 4.2.5. Natural Compounds

Some natural compounds have shown anti-inflammatory, anti-carcinogenic, anti-microbial, and antioxidant properties, so their use has been proposed for a wide variety of therapeutic applications. Unfortunately, some of these drugs are not stable in physiological media, requiring incorporation in a suitable dosage form that protects and releases them according to the needs of each treatment [151]. Using a silicone MNA to create pores in the skin structure, Paleco et al. enhanced the skin permeability of quercetin, a flavonoid which reduces the cutaneous oxidative damage induced by sunlight exposure. The level of quercetin retained in the different skin structures was evaluated by the tape-stripping technique. Microneedle pre-treatment caused a significant increase in the amount of drug released from lipid nanoparticles, in comparison with non-treated skin. Quercetin penetration into the *stratum corneum* was two-fold higher than the one obtained with the control (from 1.19 ± 0.12 mg/cm^2^ to 2.23 ± 0.54 mg/cm^2^) and penetration in the epidermis was five-fold higher (from 0.10 ± 0.01 mg/cm^2^ to 0.56 ± 0.27 mg/cm^2^) [152]. 

Dissolving maltose MNA were used by Gao et al. and Puri et al. to achieve in vitro skin permeation of honokiol and epigallocatechin-3-gallate, respectively. The first study demonstrated that it was possible to increase the delivery of honokiol contained in a propylene glycol solution by three-fold in comparison with the control (97.81 ± 18.96 μg/cm^2^ from 32.56 ± 5.67 μg/cm^2^). In addition to this, a beneficial effect was demonstrated in a psoriatic skin model, with the reduction in the release of IL-6 and the expression of Ki-67 protein, which are cell proliferation markers. Puri et al. also achieved a better delivery of epigallocatechin-3-gallate, administered in aqueous solution or carbopol hydrogel, after a pre-treatment of the porcine skin using the MNA (38.67 ± 2.96 μg/cm^2^ compared to 24.16 ± 2.11 μg/cm^2^ for control solution, 24.6 ± 2.62 μg/cm^2^ compared to 15.62 ± 0.24 μg/cm^2^) [153,154].

### 4.3. Cosmeceuticals

At present, cosmeceuticals are probably one of the most tested ingredient groups in clinical trials in combination with MNA [155]. This can be attributed mainly to the ability of MNA to deposit the active ingredient in the viable epidermis and to the simpler and faster regulatory processes associated with this field in comparison with that of medicinal products. For example, this approach is very common to treat wrinkles and two groups have already performed clinical trials using dissolving MNA for this purpose. Kim et al. organized a double-blind, randomized controlled trial with 24 women for 12 weeks, divided into two groups [156]. One group was treated daily with retinyl-retinoate MNA on the left eye crow’s feet area, and the second group was also treated daily in the same area of the right eye with ascorbic acid MNA. Before starting the trial, the drug diffusion and MNA dissolution profiles were assessed with a Franz cell diffusion setup. The results showed complete MNA dissolution after 6 h of application and both types of MNA demonstrated statistically significant differences between the initial and final state of the treated area in skin roughness, maximum roughness, average roughness, smoothness depth, and arithmetic average roughness, as measured by a skin visiometer. Skin roughness was further improved using retinyl-retinoate MNA, while ascorbic acid MNA provide better results in smoothness depth. Lee et al. performed a similar study with 21 volunteers for 4 weeks [157]. Skin elasticity, dermal elasticity, dermal density, and skin moisture content were used to evaluate the skin barrier restoration and moisturizing properties of horse oil-loaded dissolving HA MNA. Subjects applied the MNA once every two days for the length of the study and tests to check skin elasticity, dermal elasticity, dermal density, and skin moisture content were performed before, 2 and 4 weeks after patch application using a corneometer, a cutometer, and a skin scanner. Significant differences were observed in all parameters between 0 and 4 weeks, evidencing an improvement of skin state and reduction of the wrinkles. 

Collagen is another molecule used for the same objective, to replenish the protein levels lost due to natural skin aging. Sun et al. used a dissolving PVP MNA to administer type 1 collagen-rhodamine B isocyanate in human and porcine skin [100]. Fluorescence analysis showed an optimal delivery of collagen in both types of skin and an ELISA assay confirmed a non-significant decrease of functional collagen after the MNA preparation and electron beam sterilization.

MNA devices have recently been used in wound healing models to deliver antioxidants, antibacterial, and angiogenic drugs. Park et al. designed a HA dissolving MNA containing a green tea extract [158]. The main components present in this extract are polyphenols and catechins, which have previously shown inhibitory effects against Gram-positive and Gram-negative bacteria. The release rate of compounds was relatively high in the first few hours and decreased over time, being sustained for 72 h approximately. In vitro assays were performed to check the cytotoxicity and antimicrobial properties. MNA were not cytotoxic to CHO-K1, 293T, and C2C12 cells and caused a 95% reduction of the growth of Gram-positive (*Escherichia coli*, *Pseudomonas putida,* and *Salmonella typhimurium*) and Gram-negative bacteria (*Staphylococcus aureus* and *Bacillus subtilis*). Furthermore, a *Pseudomonas putida* infected wound healing model in rats was used to determine the number of colony-forming units recoverable from the wounds. The results showed that wound healing was accelerated by the use of MNA, whereas the number of bacteria recovered from the wounds was considerably reduced, from 6.18 ± 0.54 log10 in non-treated group to 2.03 ± 0.10 log10 in the group treated with MNA. 

Chi et al. obtained similar results using chitosan-dissolving MNA loaded with vascular endothelial growth factor (VEGF) at different concentrations (1–5%). The inhibition of bacterial growth (*Staphylococcus aureus* and *Escherichia coli*) was attributed to the antimicrobial properties of chitosan, since the MNA with a concentration of 4% reduced bacterial survival to almost 1%, while the control group did not show any type of inhibition. An in vivo rat model was used to check the wound healing process, and daily inspection showed that the wound closure rate was faster in the VEGF-MNA treated group than in the control and blank-MNA groups. As inflammation could reflect the infection level and the early stage of the healing process, the expression of two typical proinflammatory factors (IL-6 and TNF-α) was analyzed by immunohistochemistry at the end of the experiment. IL-6 and TNF-α were markedly expressed in tissue sections of wounds from the control group, indicating a serious inflammatory response in the wound site, while MNA treated wounds showed lower expression of both cytokines [159].

## 5. Use of MNA in Other Organs, Stimuli Responsive MNA, and Delivery of Cells

According to the results reported by a broad range of research groups, MNA seem to be a good platform to deliver drugs into the skin, for both local and systemic treatments. Furthermore, a new window was opened when microneedle technology was applied to other fields of pharmacy and medicine, in organs or tissues different from the skin. Bhatnagar et al. were successful in using dissolving PVP/PVA MNA which contained besifloxacin to overcome barriers to absorption through the cornea, which limit the treatment of ocular infections, such as diffusion through stroma, nasolacrimal drainage or tear turnover [160]. These MNA improved significantly, in an in vitro and an ex vivo model, the deposition, permeation, and antibacterial activity of besifloxacin in comparison with a drug solution applied as drops. In addition, MNA provide other interesting benefits when they are applied to the eye. Due to the curvature of the cornea, each needle of the polymeric MNA could penetrate in different ways and depths into the eye (Figure 8). This could provide a complete release in all corneal structure allowing higher activity against pathogenic agents. Likewise, by the quick dissolution of MNA, the base plate of the array structure would be slid-off with blinking and lacrimation, avoiding any invasive removal procedures. Positive outcomes were obtained in the delivery of different drugs in other eye structures, such as the sclera and the suprachoroidal area [161,162].

Another possible target tissue for microneedle-based drug delivery could be the oral mucosa. Several studies suggest that the oral mucosa is a very interesting structure for new vaccination approaches [163,164], as it elicits cellular and humoral responses comparable to other routes, enhances levels of antibodies locally and at distal sites, is easily accessible, contains lymphoid tissue, and has a different phenotype and function of cell populations. On one hand, the population of oral mucosal Langerhans cells highly expressing MHC and co-stimulatory molecules is larger than that of dermal Langerhans cells. On the other hand, oral mucosal Langerhans cells induce a stronger mixed lymphocyte response compared to skin Langerhans cells due in part to a lack of secreted suppressive soluble factors [165]. For these reasons, oral mucosal vaccination could lead to robust and effective immune responses. However, it has been underutilized in comparison to other routes due to the lack of optimal dosage forms to overcome physical barriers of the oral cavity such as salivary flow [166]. Ma et al. revealed a promising approach of using OVA and HIV antigens loaded in CMC-Lutrol F68NF^®^ coated MNA for oral vaccination in an in vivo rabbit model [167]. This study concluded that there were no significant differences between the immune response elicited by the delivery of OVA and plasmid-DNA HIV antigens in the lip or in the oral mucosa. In addition, only the microneedle-based oral cavity vaccination group stimulated a significantly higher antigen-specific IgA response in the saliva in comparison with intramuscular injection. Local treatments in the oral cavity can also be done with MNA, as demonstrated by Seon–Woo et al. [168]. In an ex vivo assay, the authors assessed the improvement of oral mucoadhesion of a microneedle-layer and its efficiency in drug delivery in comparison to a non-microneedle layer. The adhesive strength was 1.69 Nmm/cm^2^ with microneedles and 0.41 Nmm/cm^2^ without microneedles. Besides, the oral mucosal patch with microneedles delivered 42% more drug into the oral mucosa than a normal oral mucosal patch. In addition, as attachment time increased, the amount of drug delivered was significantly greater using MNA.

The use of stimuli-responsive particles is not the only option to innovate in the field of controlled release. The polymers used to fabricate the MNA themselves can also be made sensitive to different stimuli. Duong et al. based their work in the development of a pH-sensitive copolymer which dissolves only at physiological pH to improve current prophylactic DNA vaccine delivery [169,170]. In these studies, MNA were composed by polyplexes (DNA complexed with polymers, one of them especially designed to protect the genetic material), poly I:C (adjuvant), heparin, and oligo (sulfamethazine)-b-poly (ethyleneglycol)-b-poly (beta-aminoester urethane) (triblock pH-sensitive copolymer) in a layer-by-layer assembly structure. The key point of their study is that the copolymer has cationic ternary amine structure and anionic sulfonamide groups, which provide a positive charge at pH 4.03, allowing the interaction with heparin. At pH 7.4, this charge switches to negative, disassembling the patch structure and releasing its content, due to the electrostatic repulsion between heparin and the copolymer. Cellular uptake was detected by confocal laser scanning microscopy in an in vitro transfection study with RAW 264.7 macrophages and in an in vivo mice model. Results confirmed the ability of the system to generate a specific robust immune response against the loaded antigen-encoding DNA.

Cell-based therapies have the potential to make a large contribution towards currently unmet patient needs and, thus, effective manufacture of these products is essential. Translation of cell-based therapies to the clinical environment is accompanied by challenges that will require innovative solutions [171]. Controlled and targeted delivery of a cell therapy to its site of action, without compromising cell viability and functionality, is one of these challenges and could be solved by loading them into MNA able to protect them. Gualeni et al. studied the possibility of administrating cells with MNA to treat several skin injuries (burns, ulcers, and scars) and depigmentation disorders [172]. Particularly, the objective of their study was to use hollow MNA to control and target the delivery of a cell therapy using autologous melanocytes, keratinocytes, and epidermal cell suspensions to the skin. This novel technology improves significantly the currently used methods, involving skin abrasion and dressing, associated with important levels of pain, discomfort, and risk of infection. Cell viability was studied after passage of cellular suspensions through the MNA by trypan blue staining assay and hemocytometer. Functionality after extrusion was determined by evaluating cell adhesion after 24 h and cell proliferation every 48 h. Cellular phenotype was also determined by Western blot of cell lysates or by immunofluorescence. Results indicated that cell viability was not affected when they were extruded through a 75 µm hole, but it was drastically reduced when the pore size was lower than that. The number of cells was stable after MNA administration. The phenotype was maintained after the extrusion procedure with a hole size > 75 µm. In the same way, cells maintained their functionality after being injected into the skin. Ex vivo human skin sections were treated with the MNA and the injected cells expressed the specific markers of melanocytes and/or keratinocytes, showing that cells maintained their phenotype after delivery. Skin healing was also observed under in vivo conditions. At prefixed times, insertion points performed in human volunteers were imaged using a dermatological medical scanner to evaluate the closure of the microchannels. Paths created by MNA were completely closed in a time frame between 4 and 24 h.

## 6. Safety Considerations

As with any pharmaceutical formulations, MNA must be rendered safe in order to be accepted as a true alternative to classical drug delivery technologies. Three basic undesired effects could happen after the insertion of MNA: Pain, infections, and local skin reactions (inflammation, erythema, irritation, etc.).

### 6.1. Pain

Patient’s compliance and acceptance depend mainly on the pain generated by MNA in comparison with conventional injections. MNA should produce significantly less pain than a hypodermic needle, since the needle tips in MNA do not usually reach the dermis, where nervous terminations are located. Several studies have demonstrated that pain depends on the length and number of needles in the MNA. Gill et al. observed that MNA are less painful than a 26-gauge hypodermic needle and established a correlation between the pain level and the number and length of the microneedles [173,174]. Particularly, the authors tested needles with 480, 700, 960, and 1450 μm in length, finding that the pain levels they produced varied between 5% and 37% of those obtained with the hypodermic needle. Additionally, a 10-fold increase in the number of projections produced a relatively small increase in pain, from 5% to 25% in comparison with a hypodermic-needle. Bal et al. tested the pain generated by microneedles below 550 μm in length, finding very low pain scores in all cases with no significant differences between them [175]. However, pain and skin reactions can be produced by the breakage of the microneedles during insertion, which may be possible, and is especially associated to certain materials, such as glass and ceramics [95,176,177]. 

### 6.2. Infections

Open channels created by MNA can potentially become a gate for the entry of microorganisms into the body. The development of infectious processes is directly related to disruption of the skin’s barrier function. In turn, the time that the micropores remain open depends largely on some characteristics of the MNA, such as the length of the tips. This relationship has already been stated; for example, Kalluri et al. [59] studied the in vivo kinetics of pore closure using calcein imaging, and observed that this process lasts about 15 h when exposed to the environment and up to 72 h under occlusive conditions. This period is similar to the estimated one for a hypodermic needle channel. The barrier function of the skin was also evaluated using the transepidermal water loss (TEWL) levels, indicating that the skin only needs 4 h to recover its barrier function after treatment with 550 μm long microneedles. Bal et al. [175] reported a trend that using longer microneedles resulted in a higher increase in TEWL values, since a significant difference in response was observed between needles with 400 and 200 μm in length. Besides, the study of Donnelly et al. showed that microneedle puncture resulted in significantly less microbial penetration than hypodermic needle puncture and that no microorganisms crossed the viable epidermis in microneedle-punctured skin, in contrast to needle-punctured skin. Therefore, it could be stated that the correct application of MNA to skin would not cause either local or systemic infection in normal circumstances in immune-competent patients. Particularly, excised porcine skin, Silescol^®^ membranes and radiolabeled microorganisms were used in the experiments, which showed that the number of microorganisms that crossed the membranes were an order of magnitude lower when the membranes were punctured by microneedles rather than with a 21 G hypodermic needle; and the number of microorganisms penetrating the skin beyond the *stratum corneum* was approximately an order of magnitude greater than the numbers crossing Silescol^®^ membranes in the corresponding experiment [178].

### 6.3. Biocompatibility, Immunogenicity, and Local Skin Reactions

The presence of strange objects or biological incompatibility of any component could trigger skin inflammation, irritation, or erythema during or after treatment with MNA. Bal et al. checked the possible impact of microneedle length on irritation, examining the redness of the skin, and blood flow [175]. Both methods measure erythema, one of the fundamental indicators of inflammation. Changes in redness were observed after the application of solid metallic MNA, reaching maximum irritation values after 15 min and returning to the basal values in approximately 90 min. For those MNA, an increase in length (200, 300, and 400 µm) results in an increase in redness, although only the differences between the smaller length and the rest were statistically significant. Moreover, the treatment with MNA of various lengths did not result in significant differences in blood flow. On the other hand, Vicente–Perez et al. did not find any significant evidence of an increase in sera biomarkers of inflammation and irritation (TNF-α and IL-1β) after a prolonged use of polymeric MNA [179].

Low biocompatibility or allergic responses are no frequent in microneedle-based approaches, especially with polymeric MNA since the manufacturing materials are commonly transferred from other fields of pharmaceutical technology where compatibility has been guaranteed. However, other types of MNA, such as the solid and hollow ones, might be associated with some issues. For example, stainless steel MNA can suffer corrosion over time, a feature that has been clearly improved by using titanium [180]. In addition, nickel MNA have been designed and used [181] despite the fact that nickel has shown cytotoxicity and intracellular accumulation in human HaCat keratinocytes [182].

Some studies have demonstrated that MNA do not stimulate the humoral immune system, as shown in the work of Vicente–Perez et al. [179]. Here, they used three well known sera biomarkers such as C-reactive protein, TNF-α, and IgG to monitor possible infection and inflammatory processes associated with two dissolving MNA treatment regimens, using an in vivo model. TNF-α levels were undetectable in all mice, and no statistical differences were found for the other measured biomarkers, regardless of formulation type, needle density, number of applications, and mouse gender.

## 7. Conclusions

MNA have become a resource with great presence and prospects in the field of transdermal drug delivery. Their enormous potential lies in their ability to disrupt the barrier function exerted by the skin, and by doing so in a less invasive manner than other methods. The insertion of these devices does not reach the nervous terminations of the skin, making it a practically painless delivery method. On the other hand, it has been shown that the anxiety levels caused by MNA are considerably lower than those generated by other methods such as injections, which potentially would increase compliance rates, especially in the pediatric population [183]. 

They can be used to achieve both local and systemic effects as they release the drug at the interphase of *stratum corneum* and viable epidermis. The types of molecules administered by these devices are varied, being of greatest interest those intended to exert systemic effects. For example, vaccination is a field where microneedles have great potential, since doses of antigen commonly administered are not very high. In this case, the use of MNA could be a much simpler and accessible approach to the general population, allowing mass vaccination campaigns for large populations. 

MNA are a resource that has long been studied in drug delivery research, and there are already cases of innovation within this field which broaden the horizon of interest and applicability of these tools. All these facts, together with the presumable safety of microneedle-based approaches, places these devices at the forefront of biomedical and translational research. However, the MNA field has been criticized for not moving towards a clinical path. This can be attributed to different causes, such as potential difficulties in scalability, the bench-to-bedside transfer process, or the lack of information about the real applications of this technology. Although many studies using a wide range of molecules to treat different pathologies are currently ongoing, the industrial interest on MNA is lower than we could expect, regarding the positive results achieved. In this sense, research should aim to increase the evidence of MNA utility beyond vaccination, widening the range of possible treatments and other interesting possibilities to the population. Taken together with the satisfaction of patient’s needs, the microneedle technology could become an alternative to current treatments, building a bridge between the bench and the clinic.

## Figures and Tables

**Figure 1 pharmaceutics-12-00569-f001:**
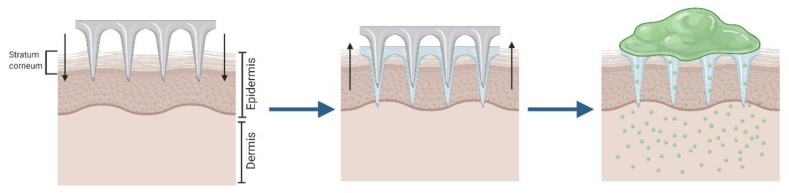
Schematic representation of the “*poke and patch”* approach with solid microneedle arrays (MNA).

**Figure 2 pharmaceutics-12-00569-f002:**
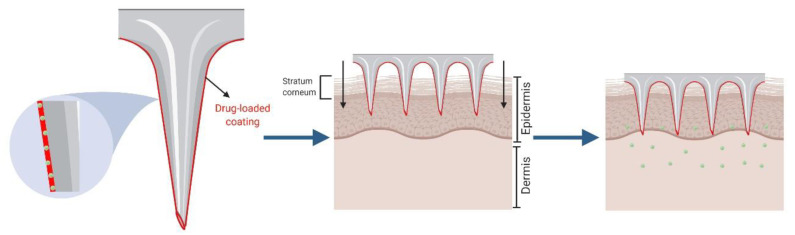
Schematic representation of *“coat and poke”* approach.

**Figure 3 pharmaceutics-12-00569-f003:**
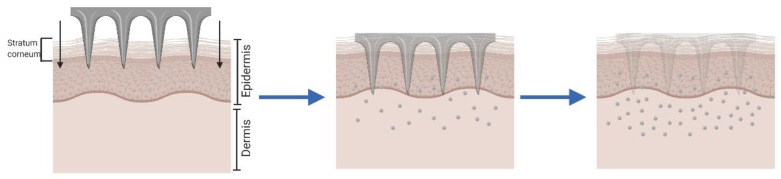
Schematic representation of *“poke and release”* approach.

**Figure 4 pharmaceutics-12-00569-f004:**
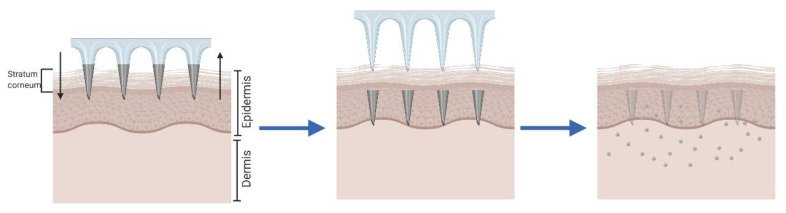
Schematic representation of rapidly separating MNA.

**Figure 5 pharmaceutics-12-00569-f005:**
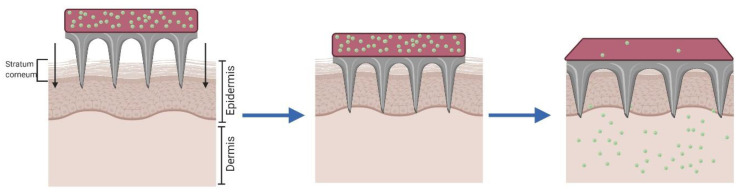
Schematic representation of hydrogel-forming or swelling MNA.

**Figure 6 pharmaceutics-12-00569-f006:**
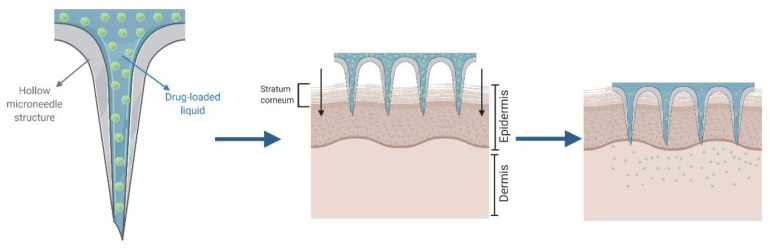
Schematic representation of *“poke and flow”* approach.

**Figure 7 pharmaceutics-12-00569-f007:**
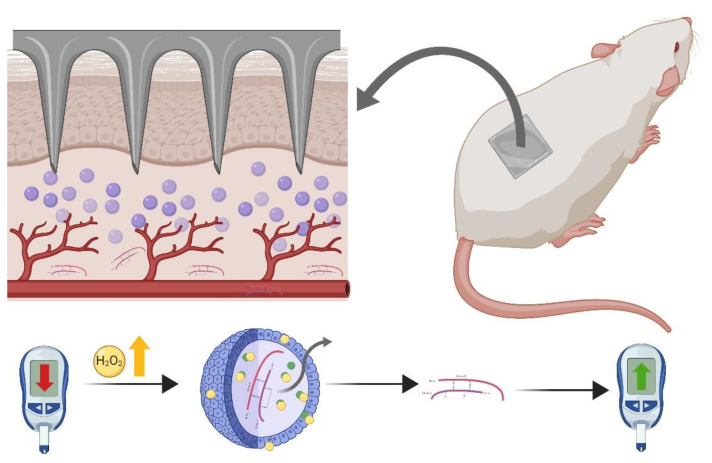
Glucose and H_2_O_2_ responsive polymeric vesicles containing insulin are loaded in polyvinylpyrrolidone (PVP)/polyvinyl alcohol (PVA) MNA, which allow an easy penetration of the vesicles directly near the dermis vessels. The constituent polymers of the vesicles are hydrolyzed at hyperglycemic states and in the presence of a H_2_O_2_ stimulus, releasing the insulin in a controlled manner. The strategy provided an effective and longer-lasting hypoglycemic effect compared to subcutaneous injections. (Graphic representation of the model used in the study by Tong’s et al. [137]).

**Figure 8 pharmaceutics-12-00569-f008:**
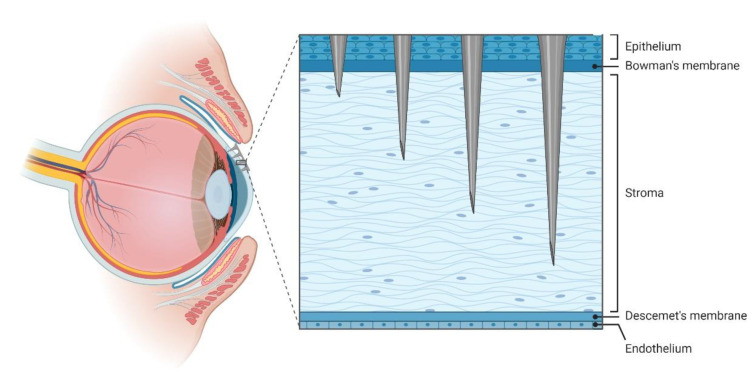
Illustration of the use of MNA on the eye. MNA can be used to deliver the drug to the anterior or posterior segments of the eye or remain in all layers of the corneal structure. The microneedles length and the physiological curvature of the cornea makes it possible to reach all structures of the tissue, allowing any type of treatment in the eye.

**Table 1 pharmaceutics-12-00569-t001:** Summary of main advantages and disadvantages of chemical and physical strategies to improve transdermal drug absorption. SC: *stratum corneum*.

Method	Advantages	Disadvantages	Ref.
Chemical enhancers	High effectiveness in combination with small molecules	Poor effectiveness in combination with macromolecules and hydrophilic molecules Inability to locate the effects on the *stratum corneum* (SC) Skin reactions (irritation, inflammation, erythema) Anti-inflammatory, anti-irritation pre-treatments are recommended	[26,27,28]
Microsystems and nanosystems	Possibility to localize the effects and drug release in the first layers of the skin	Large size can hinder the penetration of the system into the skin	[29]
Prodrugs	Improve chemical stability Avoiding skin reactions	Large size can hinder the diffusion through the skin	[30]
Iontophoresis	Rapidly responsive molecular transport Control of transport magnitude	Devices are expensive Not applicable for long periods of time due to the polarization of the skin Inability to locate the effects on the *SC* Skin reactions (irritation, inflammation, erythema)	[31,32,33,34,35]
Electroporation	Rapidly responsive molecular transport Control of transport magnitude	Devices are expensive Inability to locate the effects on the *SC* Skin reactions (irritation, inflammation, erythema)	[36,37]
Sonophoresis	Rapidly responsive molecular transport Control of transport magnitude Good effectiveness in combination with hydrophilic drugs and medium-large molecular weight	Devices are expensive Poor range of molecules administered safely Inability to locate the effects on the *SC* Skin reactions (irritation, inflammation, erythema)	[38,39,40,41]
Thermal methods	Possibility to diffuse large-size molecules	Inability to locate the effects on the *SC* Intense skin reactions (irritation, inflammation, erythema)	[42,43]
Jet injectors	Delivery of solid particles or liquids Possibility to control de depth where the drug is deposited Useful for vaccination	Not applicable for long periods of time Possibility of contamination of the devices with interstitial fluids	[44,45,46,47]

**Table 2 pharmaceutics-12-00569-t002:** Summary of advantages and disadvantages of the main materials and methods used to manufacture MNA.

Material	MNA Type	Fabrication Process	Advantages	Disadvantages	Ref.
Stainless steel, Titanium, Nickel, Gold	Solid, Hollow, Coated (array)	Laser cutting, laser ablation, etching, electropolishing, lithography, and microstereolithography	Desirable mechanical properties and high tensile strength	Fractures, corrosion, and poor biocompatibility of some metals	[78,79,80,81,82,83,84]
Alumina, Zirconia, Calcium phosphate/sulphate	Ceramic	Lithography and ceramic sintering	Good biocompatibility	Fractures	[85,86,87,88]
Silicon	Solid, Hollow, Coated (array)	Etching, Lithography	Desirable mechanical properties	High material cost, long fabrication, and fractures	[89,90,91,92]
Borosilicates (glass)	Hollow	Pulling pipettes	Good biocompatibility	Fractures	[75,93,94,95]
Sugars	Solid, Dissolving	Solvent casting or micromolding	Good biocompatibility	Mechanical properties are more difficult to achieve, stability problems, storage issues.	[96,97,98,99]
Polymers	Dissolving, Hydrogel-forming, Coated (coating)	Solvent casting or micromolding	Optimal biocompatibility, biodegradation, and absence of waste after use	Mechanical properties are more difficult to achieve	[69,100,101,102,103,104,105,106,107,108,109]

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
