# Peer review of "Microneedle-Based Delivery: An Overview of Current Applications and Trends"

_pharmaceutics, 2020, doi:10.3390/pharmaceutics12060569_

Round 1

Reviewer 1 Report

This paper reviewed overall knowledge of the current microneedle research state, from transdermal drug delivery systems to types of the microneedle, materials and manufacturing methods, various applications of the microneedle, and safety. This manuscript is written well and informative. The authors provide useful information based on the most recent research on various topics related to microneedle drug delivery using appropriate pictures and tables so that the reader can easily understand the microneedle research. Therefore, I agree that this review will be published in the Pharmaceutics after some modifications are made.

Minor comments

In the section of “2. Microneedle-based transdermal delivery application”, the classified strategies of microneedle delivery are pretty the same with the paper, Mark R. Prausnitz, Annu. Rev. Chem. Biomol. Eng. 2017. 8:177–200. Please add this reference.

Please check out some typos in the whole manuscripts.

Author Response

Response to Reviewer 1 Comments

This paper reviewed overall knowledge of the current microneedle research state, from transdermal drug delivery systems to types of the microneedle, materials and manufacturing methods, various applications of the microneedle, and safety. This manuscript is written well and informative. The authors provide useful information based on the most recent research on various topics related to microneedle drug delivery using appropriate pictures and tables so that the reader can easily understand the microneedle research. Therefore, I agree that this review will be published in the Pharmaceutics after some modifications are made.

Minor comments

Point 1: In the section of “2. Microneedle-based transdermal delivery application”, the classified strategies of microneedle delivery are pretty the same with the paper, Mark R. Prausnitz, Annu. Rev. Chem. Biomol. Eng. 2017. 8:177–200. Please add this reference.

Response 1: Thank you very much for this comment. The reference has been added in section 2, as suggested.

Point 2: Please check out some typos in the whole manuscripts

Response 2: The manuscript has been subjected to language editing; typos have been reviewed and corrected.

We thank the speaker for investing time reviewing our work and we appreciate very much the positive comments.

Reviewer 2 Report

The review article presented by Guillot et al. focus on the recent updates on the transdermal delivery application of microneedle arrays. The authors provided a comprehensive overview of various applications of microneedle and their fabrication. Overall, the article was convincing and recommended for publication.

Author Response

Response to Reviewer 2 Comments

The review article presented by Guillot et al. focus on the recent updates on the transdermal delivery application of microneedle arrays. The authors provided a comprehensive overview of various applications of microneedle and their fabrication. Overall, the article was convincing and recommended for publication.

Response: We thank the speaker for investing time reviewing our work and we appreciate very much the positive comments.

Reviewer 3 Report

Microneedle-based Delivery: an overview of current 2 applications and trends

by Antonio José Guillot et al.

This review had a general overview of the technology development of microneedle and summarized recent progress of molecules delivery, including the most recent application on the delivery of the SARS-CoV2 vaccine. Overall, the progress of this field is sufficiently reviewed. The paper is well written. Figures and Tables are well designed and organized.

Minor concerns:

  1. Some most recent research using microneedle technology to promote wound healing can be cited in the Cosmeceuticals section.
  2. On pages 20-21, 5.2 to 5.4 should be 6.2, 6.3, and 6.4.
  3. “local skin reaction” and “immunogenicity” basically are the same safety consideration. These two subsections can be combined together as Biocompatibility, Immunogenicity, or Skin irritation.

Author Response

Response to Reviewer 3 Comments

This review had a general overview of the technology development of microneedle and summarized recent progress of molecules delivery, including the most recent application on the delivery of the SARS-CoV2 vaccine. Overall, the progress of this field is sufficiently reviewed. The paper is well written. Figures and Tables are well designed and organized.

Minor concerns:

Point 1: Some most recent research using microneedle technology to promote wound healing can be cited in the Cosmeceuticals section.

Response 1: Thank you very much for this comment. We have added the following text in the Cosmeceuticals section:

MNA devices have recently been used in wound healing models to deliver antioxidant, antibacterial and angiogenic drugs. Park et al. designed a HA dissolving MNA containing a green tea extract [160]. The main components present in this extract are polyphenols and catechins, which have previously shown inhibitory effects against gram-positive and gram-negative bacteria. The release rate of compounds was relatively high in the first few hours and decreased over time, being sustained for 72h approximately. In vitro assays were performed to check the cytotoxicity and antimicrobial properties. MNA were not cytotoxic to CHO-K1, 293T, C2C12 cells and caused a 95% reduction of the growth of gram-positive (Escherichia coli, Pseudomonas putida and Salmonella typhimurium) and gram-negative bacteria (Staphylococcus aureus and Bacillus subtilis). Furthermore, a Pseudomonas putida infected wound healing model in rats was used to determine the number of colony-forming units recoverable from the wounds. The results showed that wound healing was accelerated by the use of MNA, whereas the number of bacteria recovered from the wounds was considerably reduced, from 6.18 ± 0.54 log10 in non-treated group to 2.03 ± 0.10 log10 in the group treated with MNA.

Chi et al. obtained similar results using chitosan-dissolving MNA loaded with vascular endothelial growth factor (VEGF) at different concentrations (1-5%). The inhibition of bacterial growth (Staphylococcus aureus and Escherichia coli was attributed to the antimicrobial properties of chitosan, since the MNA with a concentration of 4% reduced bacterial survival to almost 1%, while the control group did not show any type of inhibition. An in vivo rat model was used to check the wound healing process, and daily inspection showed that the wound closure rate was faster in the VEGF-MNA treated group than in the control and blank-MNA groups. As inflammation could reflect the infection level and the early stage of the healing process, the expression of two typical proinflammatory factors (IL-6 and TNF-α) was analyzed by immunohistochemistry at the end of the experiment. IL-6 and TNF-α were markedly expressed in tissue sections of wounds from the control group, indicating a serious inflammatory response in the wound site, while MNA treated wounds showed lower expression of both cytokines [161].

Point 2: On pages 20-21, 5.2 to 5.4 should be 6.2, 6.3, and 6.4.

Response 2: Thank you for noticing this mistake. The numbers have been corrected accordingly.

Point 3: “Local skin reaction” and “immunogenicity” basically are the same safety consideration. These two subsections can be combined together as Biocompatibility, Immunogenicity, or Skin irritation.

Response 3: We appreciate this comment and agree with the convenience of merging these two sections.

The new text in the manuscript is:

6.3. Biocompatibility, Immunogenicity and local skin reactions

The presence of strange objects or biological incompatibility of any component could trigger skin inflammation, irritation or erythema during or after treatment with MNA. Bal et al. checked the possible impact of microneedle length on irritation, examining the redness of the skin and blood flow [177]. Both methods measure erythema, one of the fundamental indicators of inflammation. Changes in redness were observed after the application of solid metallic MNA, reaching maximum irritation values after 15 min and returning to the basal values in approximately 90 min. For those MNA, an increase in length (200, 300 and 400 µm) results in an increase in redness, although only the differences between the smaller length and the rest were statistically significant. Moreover, the treatment with MNA of various lengths did not result in significant differences in blood flow. On the other hand, Vicente-Perez et al. did not find any significant evidence of an increase in sera biomarkers of inflammation and irritation (TNF-α and IL-1β) after a prolonged use of polymeric MNA [181].

Low biocompatibility or allergic responses are no frequent in microneedle-based approaches, especially with polymeric MNA since the manufacturing materials are commonly transferred from other fields of pharmaceutical technology where compatibility has been guaranteed. However, other types of MNA, such as the solid and hollow ones, might be associated with some issues. For example, stainless steel MNA can suffer corrosion over time, a feature that has been clearly improved by using titanium [182]. In addition, nickel MNA have been designed and used [183] despite the fact that nickel has shown cytotoxicity and intracellular accumulation in human HaCat keratinocytes [184].

Some studies have demonstrated that MNA do not stimulate the humoral immune system, as shown in the work of Vicente-Perez et al. [181]. Here, they used three well known sera biomarkers such as C-reactive protein, TNF-α and IgG to monitor possible infection and inflammatory processes associated with two dissolving MNA treatment regimens, using an in vivo model. TNF-α levels were undetectable in all mice, and no statistical differences were found for the other measured biomarkers, regardless of formulation type, needle density, number of applications and mouse gender.

We thank the speaker for investing time reviewing our work and we appreciate very much the positive comments.
